# Assessment of the Evolution of Groundwater Chemistry and Its Controlling Factors in the Huangshui River Basin of Northwestern China, Using Hydrochemistry and Multivariate Statistical Techniques

**DOI:** 10.3390/ijerph18147551

**Published:** 2021-07-15

**Authors:** Bing Zhou, Huiwei Wang, Qianqian Zhang

**Affiliations:** 1Hebei and China Geological Survey Key Laboratory of Groundwater Remediation, Institute of Hydrogeology and Environmental Geology, Chinese Academy of Geological Sciences, Shijiazhuang 050061, China; zhoubing@mail.cgs.gov.cn (B.Z.); whuiwei@mail.cgs.gov.cn (H.W.); 2China University of Geosciences, Beijing 100083, China; 3Key Laboratory of Cenozoic Geology and Environment, Institute of Geology and Geophysics, Chinese Academy of Sciences, Beijing 100029, China

**Keywords:** groundwater, hydrochemical characteristics, controlling factors, human activities, multivariate statistical techniques

## Abstract

Groundwater is an eco-environmental factor and critical resource required for human life and socioeconomic development. Understanding the evolution of groundwater chemistry and its controlling factors are imperative for preventing its deterioration and ensuring its sustainable use. We studied the characteristics of groundwater chemistry in the Huangshui River Basin in Qinghai Province, China using hydrochemical techniques. Additionally, we identified the controlling factors of groundwater chemistry in this region using multivariate statistical techniques. Seventeen hydrochemical parameters of groundwater were investigated at 156 sites in June 2019. The results showed that total hardness, Fe, NO_3_^−^, SO_4_^2−^, and Cl^−^ were primary pollution factors of groundwater in this region, and that 33.3%, 35.3%, 8.97%, 23.1%, and 7.69% of the samples exceeded Grade III standards for groundwater quality in China, respectively. Land use types also significantly affected groundwater hydrochemistry. The hydrochemical composition of groundwater in industrial areas is more strongly influenced by human activities. The major hydrochemical types identified in the region were HCO_3_-Ca·Mg and HCO_3_·SO_4_-Ca·Mg. Additionally, high proportions of SO_4_ (50.6%), Na (32.1%), and Cl (13.5%) groundwater types revealed the influence of anthropogenic activities on the groundwater hydrochemistry. Rock weathering was the major factor influencing the groundwater hydrochemistry, while evaporation–condensation and anthropogenic activities also influenced the hydrochemical characteristics of groundwater. The hydrochemical composition of groundwater was mainly controlled by silicate rock weathering. The main controlling factors of groundwater hydrochemistry were water–rock interactions, “physicochemical” factors (nature processes), domestic sewage, chemical fertilizer, and industrial sewage (human activities).

## 1. Introduction

Groundwater is an eco-environmental factor and a critical resource required for human life and socioeconomic development [1]. In general, groundwater is an important source of drinking water for human beings [2], especially in arid regions. Understanding the characteristics and controlling factors of groundwater hydrochemistry is important for preventing the deterioration of water quality and ensuring sustainable groundwater use [3].

The characteristics of groundwater chemistry are closely linked to natural factors, such as hydrogeological conditions, lithology of the vadose zone, water–rock interactions, and seawater intrusion [4]. In general, water–rock interactions lead to the dissolution of various minerals, inducing changes in the groundwater chemistry [5]. In coastal areas, seawater intrusion is usually the driving factor behind changes of groundwater chemistry [6]. Anthropogenic activity is also a major factor controlling the hydrochemical characteristics of groundwater. Previous studies have found that domestic sewage, industrial wastewater, and chemical fertilizers are important factors controlling the hydrochemical characteristics of groundwater in areas where human activity is intense, such as alluvial-pluvial fan zones and urban areas [4,7].

How to accurately identify the controlling factors of groundwater chemistry is a key topic in the hydrogeochemistry field. Traditional research methods, such as hydrochemical analyses, descriptive statistics, and identifying the proportional coefficient of hydrochemical ions, are inadequate for meeting the needs of managers. Multivariate statistical techniques have been widely used in the water environment [8,9,10,11], and have contributed to accurately identify the characteristics and control factors of groundwater chemistry.

The Huangshui River Basin is situated in the eastern part of Qinghai Province, China. The groundwater in this region is the most primary source of drinking water. Thus, ensuring groundwater quality is critical for ensuring healthy drinking water for residents and socioeconomic development of this region. In recent years, anthropogenic activities, notably rapid urbanization, industrialization, and agricultural development, have significantly affected groundwater in the Huangshui River Basin [12]. A previous study showed that the concentrations of ammonia nitrogen and total nitrogen in the Huangshui River exceeded Grade III standards for surface water quality in China because of substandard discharging of industrial and domestic sewage [13]. However, to date, there is no research on the chemical characteristics and controlling factors of groundwater in the Huangshui River Basin. Therefore, it is still difficult to implement plans to promote the sustainable use of groundwater and control the further deterioration of groundwater quality. This study aims to address this gap by providing essential inputs for groundwater management strategies and effective environmental monitoring.

In this study, we mainly determined the characteristics of groundwater chemistry in an arid region and identified the primary factors influencing groundwater chemistry using multivariate statistical techniques. These results may advance understanding of the hydrochemical evolution of groundwater in an arid region and provide essential guidance for the sustainable use of groundwater.

## 2. Materials and Methods

### 2.1. Study Area

The Huangshui River originates from Baotu Mountain in Haiyan County and flows down through Xining, the capital of Qinghai Province, Dajiachuan, and Lanzhou in Gansu Province, and finally flows into the Yellow River. The Huangshui Riveris 349 km long with an area of approximately 3200 km^2^. The Huangshui River is the third largest tributary of the Yellow River, with an annual runoff of 4.63 billion m^3^. The rivers in the Huangshui River basin mainly include the Beichuan River, the Baoku River, the Heilin River, the Xichuan River, and the Datong River (Figure 1). The study region has an arid continental climate with plateau characteristics. The average annual temperature in this region is 5.9 °C, the average value of annual precipitation is 368 mm, with main precipitation occurring between May and September [14], and the annual average evaporation is 1677 mm.

There are five types of groundwater in this area: (1) loose stratum pore water; (2) water in clastic rock fissures; (3) water in karst carbonate; (4) water in bedrock fissures; and (5) frozen water layers. Loose stratum pore water is mainly distributed in the river valley and hilly areas. Loose stratum pore water can be divided into river valley phreatic water and Loess bottom gravel phreatic water. The surface layer of the valley is mainly composed of silty sand or sub-sandy soil. Beneath the surface layer of the valley is alluvial and pluvial sandy cobble of the Holocene series, with a thickness of 5–15 m, and below it is the argillaceous sand and pebbles of the upper Pleistocene glacial moraine, with a thickness of about 20–40 m (Figure 2). The Holocene alluvium and the upper Pleistocene glacial moraine have no water-resisting layer, and the hydraulic connection is good. The Loess subsoil gravel phreatic water mainly exists in the sand gravel and argillaceous sand gravel layers deposited by glacial water of Middle Pleistocene, which have a thickness of 3–6 m. Water in clastic rock fissures is mainly found in the aquifer formed by Mesozoic and Cenozoic clastic rock layers, and the lithology of aquifer comprises sandstone, glutenite, and marl. Aquifers are mostly confined, and groundwater is mostly brackish water or salt water. The buried depth of the aquifer floor is 74.6–200 m. Karst water located in carbonate crannies is distributed in the Laoye mountain area, and the lithology of aquifer is Cambrian dolomite, limestone, and crystalline limestone along with plagioclase basalt and siliceous rock. The bedrock fissures water is mainly distributed in the northern Bedrock mountains area, and the groundwater is mainly distributed in metamorphic rock fissures and the structural belt of the Proterozoic–Lower Proterozoic strata. The layer of frozen water is mainly distributed in a permafrost area in the northwest of the basin center (over 3800 m above sea level), and the thickness of the aquifer is less than 5 m.

The main modes of groundwater compensation are river infiltration and groundwater runoff recharge in the upper reaches of the river, followed by precipitation, return of agricultural irrigation water, and seepage from canals. Groundwater drainage occurs primarily via manual exploitation. There are frequent exchanges between groundwater and surface water. The upper reaches of the study area, after the Baoku River and the Heilin River, have passed through the outlet of the mountain, and the river water replenished groundwater. As the two rivers converge, the valley begins to narrow and the basement rises, which lead to groundwater flowing out as springs replenish the river. The river flows through the outlet of the Laoye mountain, the valley suddenly widens, the river flow rate decreases, and the surface water begins to recharge the groundwater (Figure 1).

### 2.2. Groundwater Sampling and Laboratory Analyses

Overall, 156 samples were collected in June 2019 (Figure 1). All wells selected for groundwater sampling are mainly used for drinking and agricultural irrigation purpose. The average distance to the groundwater table in these wells was 10.5 m (ranging between 3.0 m and 45.0 m). The wells were purged for 5 min before the samples were collected. The dissolved oxygen (DO) and pH values of the samples were measured in the site using an HQ40D multiparameter instrument (Hach, Loveland, CO, USA).

All samples were stored in two plastic sampling bottles of different sizes (500 mL and 1.5 L) and were used to analyze the water quality parameters. Samples that were not preprocessed were used for analyses of anions, whereas samples used for cation analyses were acidified to a pH below 2. The hydrochemical indicators were analyzed in the laboratory at the Groundwater Mineral Water and Environmental Monitoring Center of the Chinese Academy of Geological Sciences. Details of the analytical parameters, methods, and detection limits are presented in Table 1.

### 2.3. Data Analysis

Spearman correlation analysis was performed to determine significant relationships between groundwater indicators, while a principal component analysis (PCA) was conducted to distinguish the primary factors influencing groundwater chemistry. All statistical tests were performed on the R computing platform, version 4.0.1 (New Zealand) and with the SPSS software package, version 21.0 (SPSS Inc., Chicago, IL, USA).

## 3. Results and Discussion

### 3.1. Characteristics of Hydrochemical Parameters in Groundwater

The characteristics of groundwater hydrochemical parameters are summarized in Table 2. The CO_3_^2−^ concentrations in the groundwater samples were all below the detection limit. In this area, pH values of groundwater samples were from neutral to weakly alkaline (6.52–9.65), with an average value of 7.54. A total of 1.28% of the samples had pH values that exceeded the standard for drinking water in China [15]. The aquifer is in an overall oxidizing environment, and the mean concentrations of DO is 5.75 mg/L (ranging from 0.940 mg/L to 14.8 mg/L). The predominant cation array was expressed as Na^+^ > Ca^2+^ > Mg^2+^ > K^+^, while the anion array was expressed as HCO_3_^−^ > SO_4_^2−^ > Cl^−^ > NO_3_^−^ >NO_2_^−^. It is striking that NO_3_^−^, SO_4_^2−^, and Cl^−^ concentrations in 8.97%, 23.1%, and 7.69% of the respective samples, respectively, exceeded the standards for drinking water in China [15]. These results showed that the groundwater has been influenced by anthropogenic activities, such as sewage and chemical fertilizers [16]. The mean concentration of TDS was 888 mg/L, and 22.4% of the samples exceeded the concentration specified for the standard of drinking water in China [15]. In addition, TH and Fe concentrations occurred in the ranges of 10.0 mg/L–2572 mg/L and 0.01 mg/L–130 mg/L, respectively, with mean concentrations of 453 mg/L and 2.41 mg/L, respectively. The exceeding standard rates of TH and Fe were 33.3% and 35.3%, respectively, indicating that they were significantly associated with groundwater pollution in the Huangshui River Basin. In general, high TH values in drinking water may induce a variety of diseases, such as vascular disease and acute myocardial infarction [17,18]. High concentration of Fe in surface or groundwater can lead to chronic poisoning and endanger to human health, such as HFE-associated hereditary hemochromatosis and β-thalassemia [19]. Therefore, local environmental managers should pay close attention to these parameters to ensure safe drinking water for local residents.

### 3.2. Effect of Land Use on the Groundwater Chemistry

Among the hydrochemical parameters, pH, Cl^−^, SO_4_^2−^, NO_3_^−^, TDS, and Fe were selected to analyze the effect of land use on the groundwater chemistry. As shown in Figure 3, the values of six hydrochemical parameters were the lowest in the forest area due to the groundwater being less affected by human activity in this area. The concentration of Cl^−^, NO_3_^−^, Fe, and pH value were higher in the industrial area than in the other areas (Figure 3a,b,d,f). This is mainly because the groundwater in the industrial area was affected by various pollutants from industrial wastewater, domestic sewage, waste materials from wear of machinery, and vehicles, etc. During the investigation and sampling, we found that there are several metal factories in the industrial park area and that the industrial wastewater is directly discharged into the Beichuan River. In addition, the concentrations of SO_4_^2−^ and NO_3_^−^ were high in the village area, which indicated that the groundwater may be affected by domestic sewage [2]. Indeed, in the village area, the domestic sewage was often discharged by digging deep wells. This, in turn, leads to an increase in the concentration of SO_4_^2−^ and NO_3_^−^ in groundwater.

### 3.3. Hydrochemical Types

The piper diagram is often used to show hydrochemical types of groundwater [5,20]. In this region, the primary hydrochemical types of groundwater were HCO_3_-Ca·Mg (42%) and HCO_3_·SO_4_-Ca·Mg (22%) (Figure 4), reflecting freshwater recharge and carbonate mineral dissolution. In addition, SO_4_-type (51%), Na-type (40%), and Cl-type (14%) water accounted for high proportions, indicating that the groundwater chemistry have been influenced by human activity (e.g., domestic sewage, industrial wastewater, and chemical fertilizer, etc.) [4]. These results are consistent with previous studies. Researchers found that Cl^−^ and SO_4_^2−^ are important indicators that reflect the impact of human activities on groundwater chemistry [21]. Güler et al. [22] found that human activities have an important impact on groundwater chemistry.

Land use types reflect human activity directly and are closely related to groundwater chemical compositions [23]. As shown in Table 3 and Figure 5, the main hydrochemical type of groundwater in the forest region was HCO_3_-Ca·Mg, and the proportions of SO_4_ (35%), Na (18%), and Cl (0%) were lower than other land use types. By contrast, the main hydrochemical types of groundwater in urban and industrial areas were SO_4_, Na, and Cl types, reflecting the influence of intense human activities (such as domestic sewage and industrial wastewater) on the groundwater chemistry in these areas.

As shown in Table 4, the main hydrochemical types of loose stratum pore water and clastic rock fissures water were HCO_3_-Ca·Mg. However, the proportions of SO_4_, Na, and Cl types were higher in loose stratum pore water than in clastic rock fissures water. This may be due to the groundwater table being deeper in the clastic rock fissures water than in the loose stratum pore water. Thus, the effects of human activity on clastic rock fissures water were relatively less significant [10].

### 3.4. Controlling Factors of Groundwater Hydrochemical Evolution

Gibbs plots could explain the effect of atmospheric precipitation, rock weathering, and evaporation on the chemical compositions of groundwater [24]. From Figure 6, most samples were concentrated in fields where rock weathering was a dominant feature, indicating that rock weathering is a major factor influencing the groundwater hydrochemical evolution in the Huangshui River basin. A few samples tended to shift to the upper right side of the figure, indicating that evaporation–condensation also affects the hydrochemical composition of groundwater. However, these explanations do not mean that groundwater hydrochemical evolution is not affected by anthropogenic activities. The effects of anthropogenic activities on groundwater hydrochemistry are discussed below.

### 3.5. Water-Rock Interactions (Mineral Dissolution and Ions Sources)

The bivariate cross plotting can be useful to identify the sources of major ions [25]. Previous studies found that the equimolar ratio of Cl^−^/Na^+^ was able to identify the dominance of halite dissolution for sodium in groundwater [26]. From Figure 7a, most Cl^−^/Na^+^ values were lower than 1 (there were only 6 sites of the ratios were greater than 1), which indicated that Na^+^ could come from other sources such as silicate weathering or cation exchange. The equimolar ratio of (Ca^2+^ + Mg^2+^)/(HCO_3_^−^ + SO_4_^2−^) can identify the source of Ca^2+^, Mg^2+^, and SO_4_^2−^. Previous studies have shown that: if the ratio is more than 1, it was mainly from the dissolution of carbonate; if the ratio is less than 1, it mainly originated from the dissolution of silicate; and, if the ratio is close to 1, it was from both the dissolution of carbonate and the dissolution of silicate [27]. As shown in Figure 7b, the ratio of most samples was close to the 1:1 line, demonstrating that Ca^2+^, Mg^2+^, and SO_4_^2−^ originate from the dissolution of carbonate and silicate. In addition, the ratio endmember plot of Ca^2+^ /Na^+^ VS Mg^2+^/Na^+^ could further distinguish the ions of groundwater from carbonate rock weathering, silicate rock weathering, or evaporite rock weathering (Figure 7c). As can be seen from Figure 7c, most samples were mainly concentrated between the control area of silicate and carbonate rocks, indicating that the ions of groundwater were controlled by silicate rocks and carbonate weathering and dissolution, and that the ions were mainly controlled by silicate rock weathering.

### 3.6. Correlations between the Hydrochemical Parameters

The correlations between the chemical components could reflect the source of each parameter [7,28]. As it can be seen from Figure 8, there were significant positive correlations between TH, TDS, Na^+^, Ca^2+^, Mg^2+^, Cl^−^, HCO_3_^−^, and SO_4_^2−^. Thus, these chemical parameters of groundwater evidently originated from the same source, for example, rock weathering [29]. In addition, there was a strong positive correlation between COD, Fe, and Mn, indicating their common source, such as industrial wastewater. However, NO_3_^−^ and NO_2_^−^ were not significantly correlated with other indicators. Therefore, NO_3_^−^ and NO_2_^−^ may come from different sources with other hydrochemical parameters, such as chemical fertilizer or soil nitrogen.

### 3.7. Identifying the Primary Factors Influencing Groundwater Chemical Characteristics

The PCA can identify similar relationships in groundwater parameters; therefore, it is used to distinguish controlling factors that affect groundwater chemistry. In this study, 14 hydrochemical parameters were selected to perform a PCA to identify major controlling factors of groundwater chemistry in the Huangshui River Basin (Table 5). Considering that the eigenvalue is greater than 1, we identified three PCs (controlling factors of groundwater chemistry) using the PCA method, and determined that the cumulative variance was 75.7%.

Factor 1 explained 48.7% of the total variance and demonstrated strong positive loadings for TDS, TH, SO_4_^2−^, Ca^2+^, Na^+^, HCO_3_^−^, Cl^−^, and Mg^2+^, and a moderate positive loading for NO_3_^−^, revealing the impacts of anthropogenic activity and natural processes [6,10]. As noted in Section 3.3, rock weathering is a major factor influencing groundwater hydrochemistry in this region. Therefore, high concentrations of TDS, TH, Na^+^, HCO_3_^−^, Mg^2+^, and Ca^2+^ are closely correlated with water–rock interactions. Previous studies found that Cl^−^, SO_4_^2−^, and NO_3_^−^ could reflect the effects of anthropogenic activities on groundwater chemistry, such as domestic sewage and chemical fertilizers [15,21]. During the investigation of the study area, we found that rural residents were accustomed to digging wells to discharge domestic sewage, which could easily lead to increases of Cl^−^, SO_4_^2−^, and NO_3_^−^ concentrations in groundwater [30]. Furthermore, nitrogen fertilizers (e.g., urea and ammonium chloride) and compound fertilizers are the principal agricultural inputs used in the region. Thus, the high levels of Cl^−^, SO_4_^2−^, and NO_3_^−^ in groundwater were closely associated with domestic sewage and agricultural fertilizers. In sum, PC1 is considered to denote the water–rock interactions along with domestic sewage and chemical fertilizers.

Factor 2 explained 16.9% of the total variance and had strong positive loadings for Mn and Fe, and a moderate positive loading for COD. Previous studies found that the high concentration of Fe and Mn in groundwater could reflect that the aquifer is in a reductive environment [31] and that the groundwater has been affected by industrial effluents [10]. In the Huangshui River basin, the mean DO value of groundwater was 5.75 mg/L, which implies an oxidizing groundwater environment. Therefore, higher Fe and Mn concentrations in groundwater were likely to originate from industrial wastewater, especially from steel processing plants located in the city of Xining. Indeed, the highest mean concentrations of Fe (11.4 mg/L) and Mn (0.531 mg/L) in groundwater were recorded for samples obtained from the industrial area, which further supports our conclusion. In general, COD is an indicator of organic contaminants and mainly derives from point sources, such as discharge of wastewater treatment plants and industrial effluents [32]. Considering that COD, Fe, and Mn are classified as one kind of control factors, and that PC2 and PC1 are independent of each other, COD in groundwater therefore mainly comes from industrial sewage. Thus, the second factor (PC2) that controls groundwater in this area is industrial wastewater.

Factor 3 explained 16.9% of the total variance and had strong positive loading for pH and a slight positive loading for NO_2_^−^. The pH index is comprehensive and reflects the acid–alkaline nature of groundwater. However, it cannot indicate a pollution source. Therefore, some scholars define pH as a “physicochemical” source [33,34] because it is influenced by a combination of physics and the chemical properties of water. Therefore, PC3 refers to “physicochemical” factor.

## 4. Suggestions for Environmental Management

Identification of the characteristics and controlling factors of groundwater chemistry is necessary to ensure the sustainable use of water resources and to control further deterioration of water quality. We found that the hydrochemical properties of groundwater in this area have been affected by anthropogenic activities, due to the concentrations of Cl^−^, SO_4_^2−^, NO_3_^−^, TH, and Fe exceeding standard values. High concentrations of NO_3_^−^ in drinking water could trigger a variety of diseases, such as “blue baby syndrome” [35]. Moreover, high SO_4_^2−^ concentrations in groundwater could lead to the corrosion of plumbing systems and associated bioaccumulation in fish [36]. In addition, we identified that the primary controlling factors of groundwater chemistry are domestic sewage, industrial wastewater, and chemical fertilizer. Therefore, we recommend that the environmental protection department prohibits the digging of wells to drain wastewater in the region of villages and industrial parks. Optimized fertilization strategies should also be promoted within the agricultural sector to prevent agricultural non-point pollutants in this region from affecting the groundwater chemistry.

## 5. Conclusions

We combined hydrochemical methods and multivariate statistical techniques to identify the hydrochemical properties and controlling factors of groundwater in the Huangshui River Basin in northwestern China. Our conclusions are summarized below.

The hydrochemistry of groundwater has been influenced by human activities. The concentrations of TH, Fe, NO_3_^−^, SO_4_^2−^, and Cl^-^ in the samples exceeding the drinking water standards in China (GB/T14848-2017) were 33.3%, 35.3%, 8.97%, 23.1%, and 7.69%, respectively. The hydrochemical composition of groundwater in the industrial area is more strongly influenced by human activities. Therefore, environmental managers should pay close attention to these parameters to prevent them from posing a threat to healthy drinking water for the local population.

The main hydrochemical groundwater types were HCO_3_-Ca·Mg and HCO_3_·SO_4_-Ca·Mg. The proportions of SO_4_-type (50.6%), Na-type (32.1%), and Cl-type (13.5%) were high, which further confirmed that the groundwater chemistry has been influenced by human activity. In addition, the land use types significantly affected the hydrochemical characteristics of groundwater.

Rock weathering was the primary factor influencing the hydrochemistry of groundwater in this region. Evaporation–condensation and human activities have additional effects on the hydrochemical properties of groundwater. The hydrochemical composition of groundwater was mainly controlled by silicate rock weathering.

We recognized three factors influencing groundwater chemistry in this region based on the PCA results. These factors were water–rock interactions, domestic sewage, agricultural fertilizers, industrial sewage, and the “physicochemical” factor. Therefore, we recommend that the environment protection department prohibit the digging of wells to drain domestic and industrial sewage from villages and industrial parks. Optimized fertilization strategies should be promoted within the agricultural sector to prevent the groundwater chemistry from being affected by agricultural non-point pollutants in this region.

## Figures and Tables

**Figure 1 ijerph-18-07551-f001:**
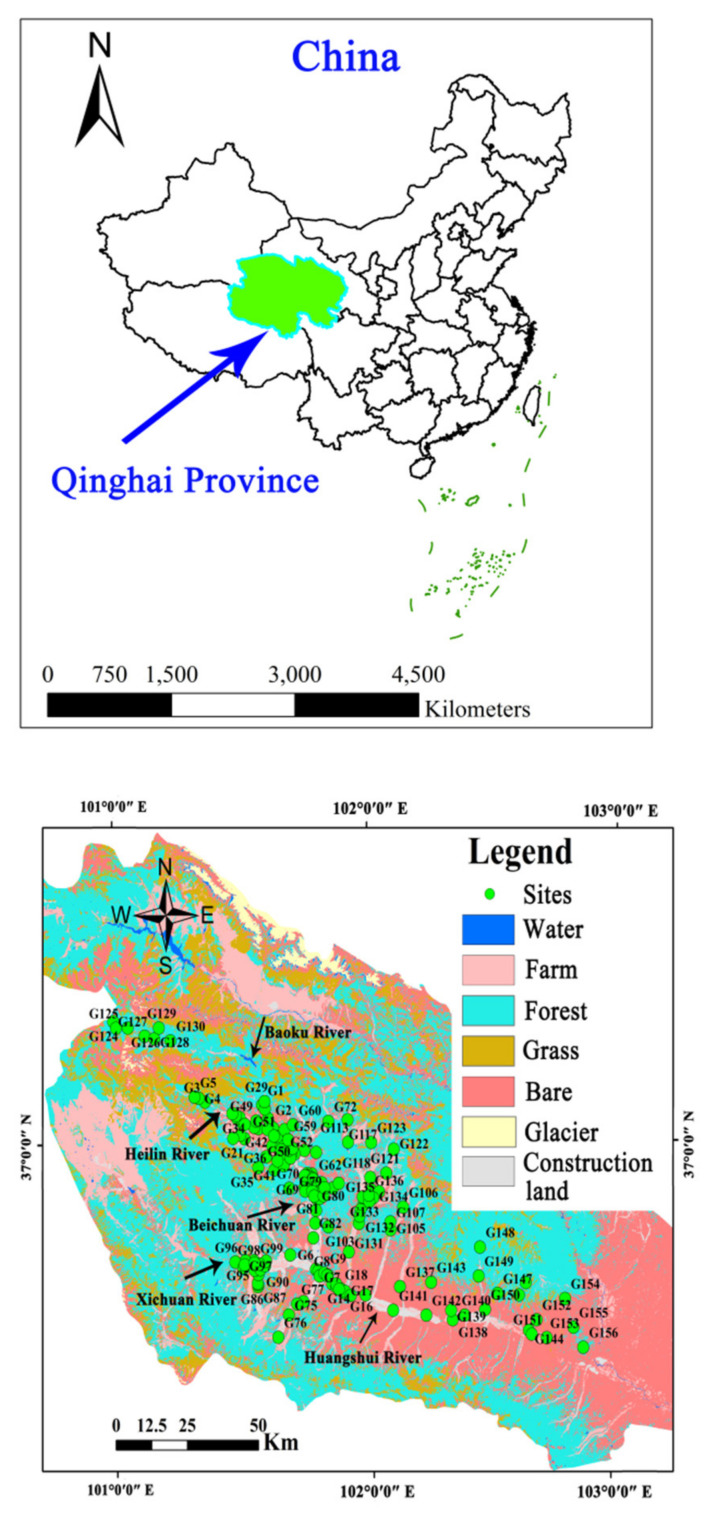
Groundwater sampling sites in the Huangshui River basin, Qinghai Province, China.

**Figure 2 ijerph-18-07551-f002:**
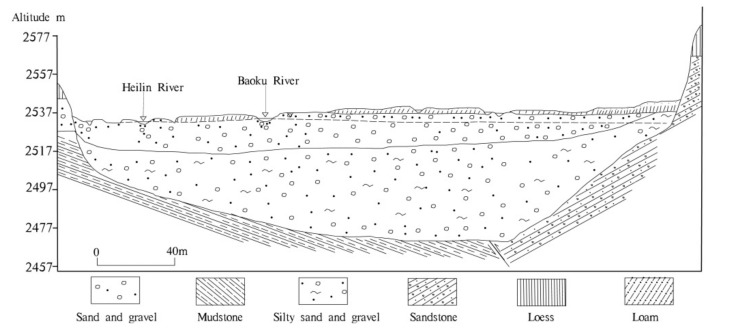
The hydrogeological cross-section of the Huangshui River basin.

**Figure 3 ijerph-18-07551-f003:**
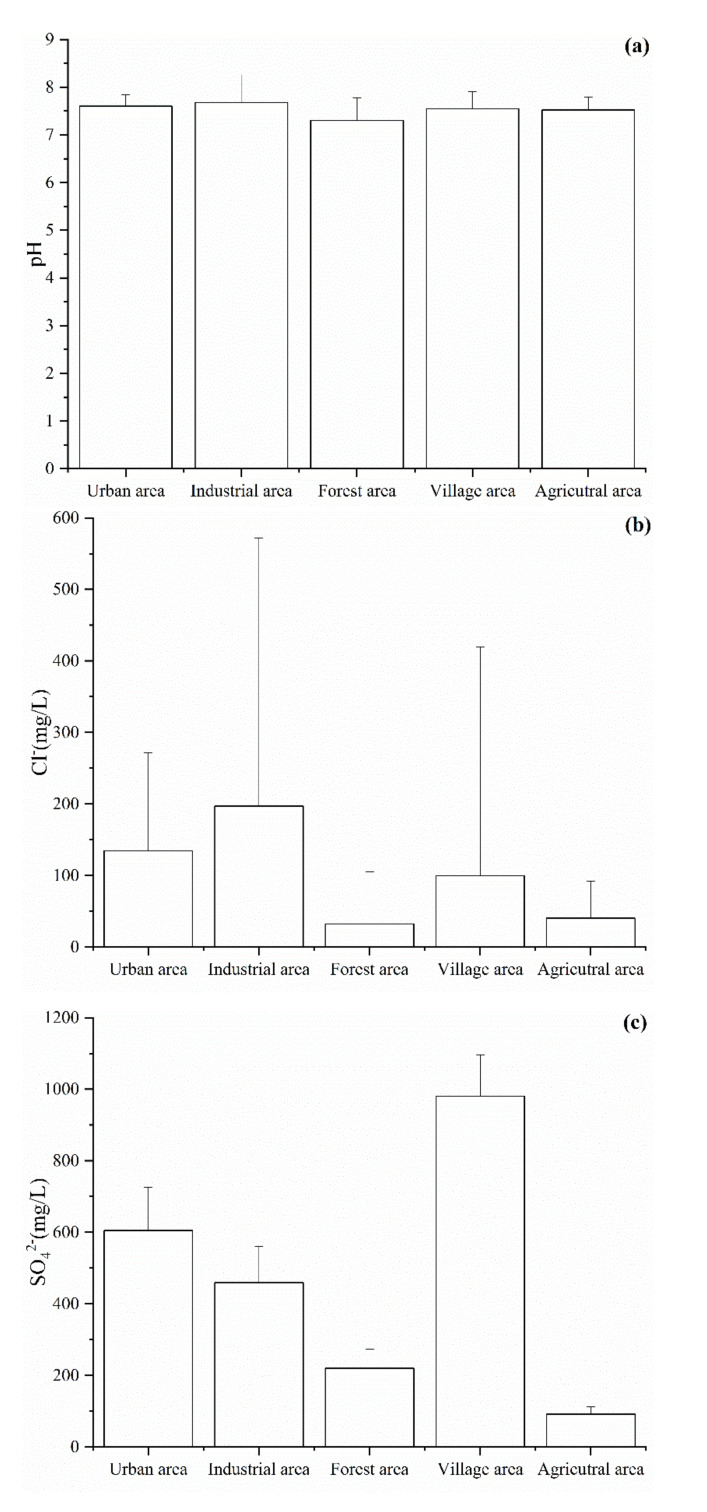
Spatial pattern of the (**a**) pH, (**b**) Cl^−^, (**c**) SO_4_^2−^, (**d**) NO_3_^−^, (**e**) TDS and (**f**) Fe values in the Huangshui River basin.

**Figure 4 ijerph-18-07551-f004:**
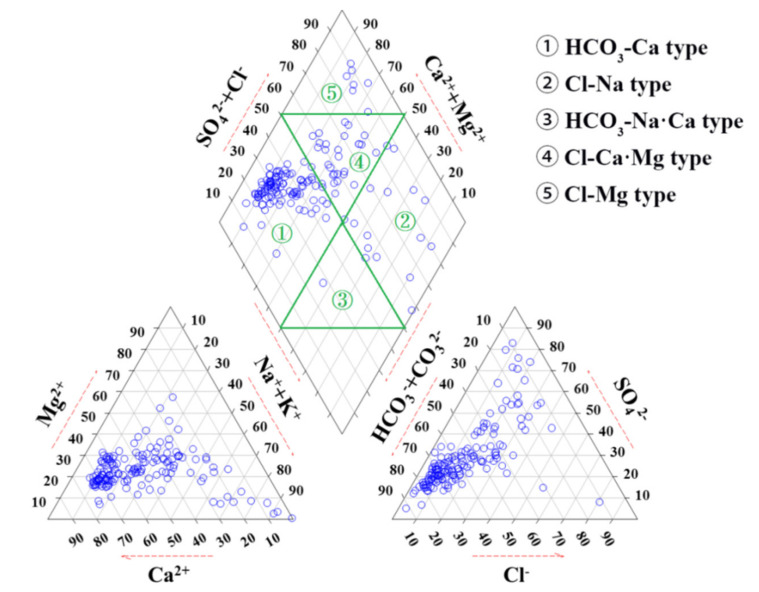
Piper diagrams of groundwater in Huangshui River basin.

**Figure 5 ijerph-18-07551-f005:**
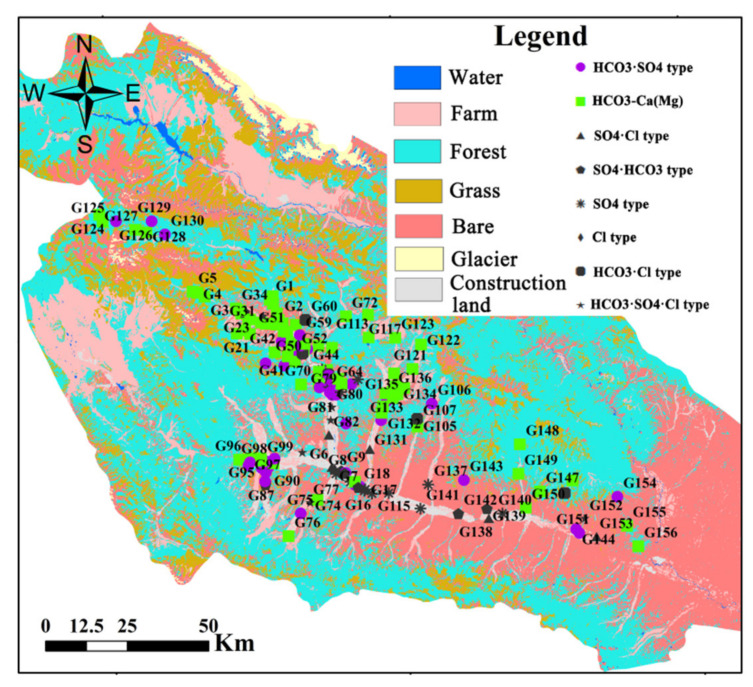
Hydrochemical type of groundwater for the sampling site in Huangshui River basin.

**Figure 6 ijerph-18-07551-f006:**
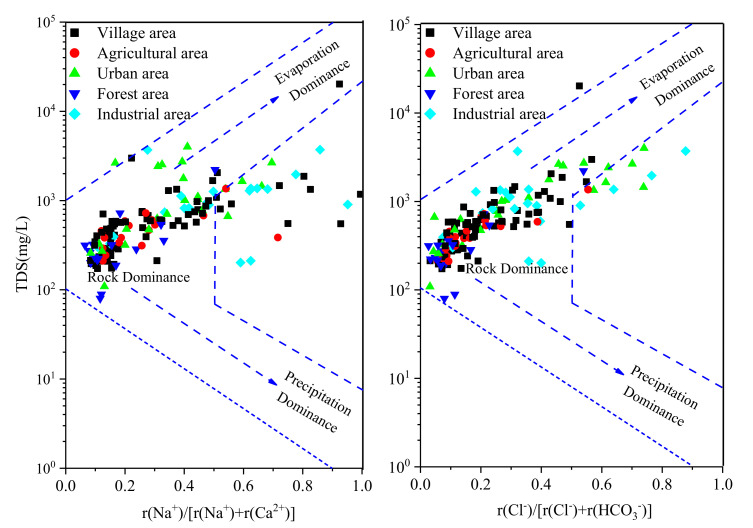
Gibbs diagram for groundwater in Huangshui River basin.

**Figure 7 ijerph-18-07551-f007:**
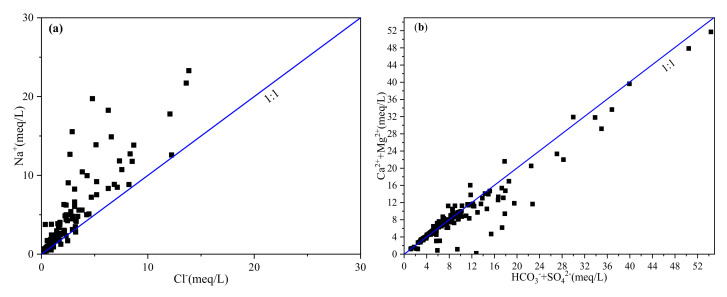
Ionic ratio plots of the major ions: (**a**) Cl^−^/Na^+^, (**b**) (Ca^2+^+Mg^2+^)/(HCO_3_^−^+SO_4_^2−^), (**c**) Ca^2+^ /Na^+^ VS Mg^2+^/Na^+^.

**Figure 8 ijerph-18-07551-f008:**
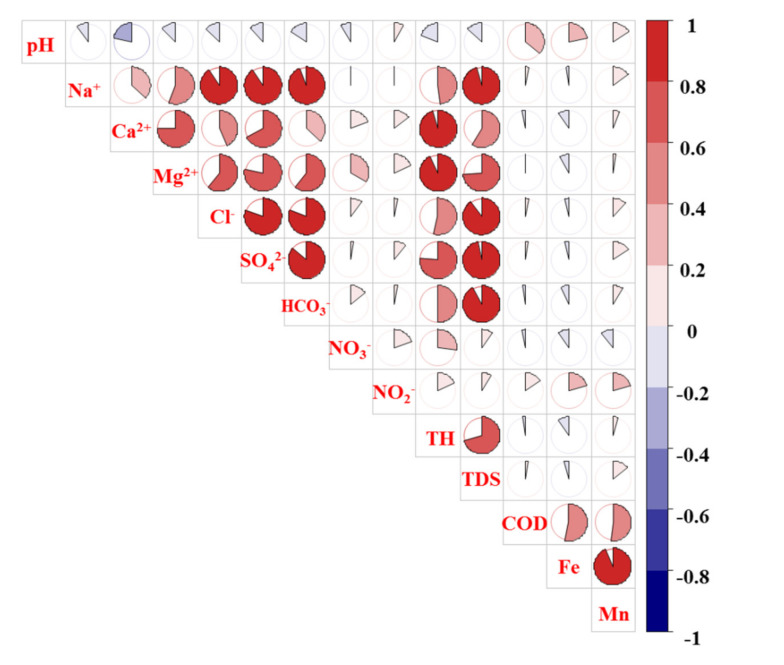
Correlation analysis of groundwater hydrochemistry component in Huangshui River basin.

**Table 1 ijerph-18-07551-t001:** Hydrochemical parameters, analytical method, equipment and detection limits.

Parameters	Analytical Method	Analytical Equipment	Detection Limit (mg/L)
Nitrate [NO_3_^−^]	Spectrophotometry	Perkin-Elmer Lambda 35, Waltham, MA, USA	0.664
Nitrite [NO_2_^−^]			0.003
Chloride [Cl^−^]			1.0
Sulfate [SO_4_^2−^]			0.75
Potassium [K^+^]	Inductively coupled plasma-mass spectrometry	Agilent 7500ce ICP-MS, Tokyo, Japan	0.05
Sodium [Na^+^]			0.01
Calcium [Ca^2+^]			4.0
Magnesium [Mg^2+^]			3.0
Iron [Fe]			0.0045
Manganese [Mn]			0.0005
Bicarbonate [HCO_3_^−^]	Acid–base titration	—	5.0
Carbonate [CO_3_^2−^]	Acid–base titration	—	5.0
Total dissolved solids [TDS]	Gravimetric methods	—	—
Chemical oxygen demand (COD)	Alkaline permanganate oxidation	Hach, USA, DRB200; Shimadzum, Japan, UV-1700	0.05
Total hardness [TH]	EDTA titration method	—	1.0

**Table 2 ijerph-18-07551-t002:** Summary of groundwater chemical indicators in Huangshui River basin.

Parameters	Min (mg/L)	Max (mg/L)	Mean (mg/L)	SD	CV (%)	Standard	Exceed Standard Rate(%)
pH	6.52	9.65	7.54	0.394	5.22	6.5–8.5	1.28
DO	0.940	14.8	5.75	2.51	43.7	—	—
K^+^	0.750	89.5	5.79	8.69	150	—	—
Na^+^	2.69	6557	133	535	403	200	14.7
Ca^2+^	2.06	603	115	100	86.8	—	—
Mg^2+^	1.17	315	40.2	47.5	118	—	—
Cl^−^	1.75	2721	103	267	259	250	7.69
SO_4_^2−^	10.0	2221	278	741	267	250	23.1
HCO_3_^−^	48.8	4210	324	336	104	—	—
NO_3_^−^	0.200	391	40.7	49.1	121	88.6	8.97
NO_2_^−^	0.002	1.40	0.040	0.146	364	3.29	0.00
TH	10.0	2572	453	418	92.1	450	33.3
TDS	80.0	20190	888	1715	193	1000	22.4
COD	0.230	15.4	1.26	1.56	123	3.00	2.56
Fe	0.010	130	2.41	13.6	565	0.3	35.3
Mn	0.005	6.01	0.151	0.660	438	0.1	13.5

Note: Min: minimum value; Max: maximum value; Mean: average value; SD: standard deviation; CV: coefficient of variation; Standard is grade Ⅲ standard for groundwater quality in China (GB/T14848-2017).

**Table 3 ijerph-18-07551-t003:** Statistic table of groundwater hydrochemistry type of different land use in this region.

Hydrochemical Type	Samples	Proportions of Hydrochemical Type (%)			
		HCO_3_-Ca(Mg) Type	SO_4_ Type	Cl Type	Na Type
Urban area	25	32	68	16	52
Industrial area	21	14	76	29	67
Forest area	17	59	35	0	18
Village area	72	49	43	14	33
Agricultural area	21	48	43	10	14
Sum	156	42	51	14	40

**Table 4 ijerph-18-07551-t004:** Statistic table of groundwater hydrochemistry type of different type groundwater in this region.

Groundwater Type	Samples	Proportions of Hydrochemical Type (%)			
		HCO_3_-Ca(Mg) Type	SO_4_ Type	Cl Type	Na Type
Loose stratum pore water	122	42	52	14	42
Clastic rock fissures water	34	59	35	12	12

**Table 5 ijerph-18-07551-t005:** Rotated component matrix of PCA on datasets of hydrochemical parameters in the Huangshui River basin.

Parameters	Factors		
	PC1	PC2	PC3
TDS	0.947	0.124	0.206
Mg^2+^	0.938	0.057	−0.123
TH	0.902	0.102	−0.287
SO_4_^2−^	0.887	0.181	0.238
Cl^−^	0.881	0.125	0.288
Na^+^	0.844	0.144	0.372
HCO_3_^−^	0.821	−0.087	0.003
Ca^2+^	0.803	0.125	−0.391
NO_3_^−^	0.631	−0.347	−0.131
Mn	0.113	0.919	0.081
Fe	−0.037	0.913	0.031
COD	0.126	0.509	0.195
pH	−0.199	−0.100	0.784
NO_2_^−^	0.209	0.469	0.488
Eigenvalue	6.81	2.37	1.41
% of Variance explained	48.7	16.9	10.1
% Cumulative Variance	48.7	65.6	75.7

## Data Availability

Not applicable.

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
