# Peer review of "Assessment of the Evolution of Groundwater Chemistry and Its Controlling Factors in the Huangshui River Basin of Northwestern China, Using Hydrochemistry and Multivariate Statistical Techniques"

_ijerph, 2021, doi:10.3390/ijerph18147551_

Round 1
Reviewer 1 Report
Avoid cutting words
Review writing
Table 3 does not show statistical data
Support the claims about the influence of anthropogenic activities in the alteration of the chemical composition of water.
There is confusion in some concepts of hydrogeochemical factors and processes.
There is a lack of clarity in the concept of pH, which they refer to in the following text.
Elaborate other hydrogeochemical graphs to evaluate the water evolution
Check grammar
Author Response
- Avoid cutting words
Response: Thank you for your reminder. We have removed the auto hyphenation from the article.
- Review writing
Response: We have checked the paper writing.
- Table 3 does not show statistical data
Response: Thank you for your suggestions. We have added the statistical data in Table 3.
- Support the claims about the influence of anthropogenic activities in the alteration of the chemical composition of water.
Response: We have added the discussion about the effect of anthropogenic activities on chemical composition of groundwater in the section 3.3.
- There is confusion in some concepts of hydrogeochemical factors and processes.
Response: We have revised “3.3 Impacts of chemical processes on groundwater” as “3.4 Controlling factors of groundwater hydrochemical evolution”. The section of “3.5. Identifying the primary factors influencing groundwater chemistry” is revised “3.7. Identifying the primary factors influencing groundwater chemical characteristics”.
- There is a lack of clarity in the concept of pH, which they refer to in the following text.
Response: We have revised “pH” as “pH values”
- Elaborate other hydrogeochemical graphs to evaluate the water evolution
Response: We have added the hydrogeochemical graphs to evaluate the groundwater evolution and sources of ions in the section 3.5.
8.Check grammar
Response: The syntax errors and language have been carefully checked and corrected by a native English speaker.
Reviewer 2 Report
This study evaluates the controlling factors of groundwater chemistry in the Huangshiu River Basin in Qinghai Province, China. The authors collected 156 samples from all the wells mainly used for drinking and agricultural irrigation purpose. I think this manuscript is well written and provides important information from an environmental perspective. However, some points are still unclear, so it should be explained and discussed more clearly. Therefore, the manuscript needs some modifications to be published. I have added some comments and suggestions attached.

Author Response
This study evaluates the controlling factors of groundwater chemistry in the Huangshui River Basin in Qinghai Province, China. The authors collected 156 samples from all the wells mainly used for drinking and agricultural irrigation purpose. I think this manuscript is well written and provides important information from an environmental perspective. However, some points are still unclear, so it should be explained and discussed more clearly. Therefore, the manuscript needs some modifications to be published. I have added some comments and suggestions attached.
- If all 156 samples would not have been collected in one day, then how was the effect of precipitation taken into account?
Response: Thank you for your comment. We were unable to complete the sampling in one day due to the large number of samples (n=156). However, the impact of rainfall on groundwater chemistry has been taken into account in the design of the sampling scheme. As mentioned earlier, evaporation is high (1677 mm) and rainfall is low (368 mm) in the study area (especially in the early rainy season), and depth of groundwater table is deeper (mean value is 10.5 m ), so it is difficult for rainfall to penetrate into aquifer in the early rainy season. Therefore, rainfall has no effect on the hydrochemistry of the groundwater during our sampling period.
- In the ‘2.1. Study area’ part: The authors said that “There are frequent exchange between groundwater and surface water, with river water recharging groundwater in upstream of the Huangshui River, while groundwater is gradually discharged to recharge the other area of the river.” Could you please provide more quantitative values with references?
Response: Thank you for your suggestions. In this study area, the research degree of hydrogeology is relatively low, therefore, we have not found the quantitative value about surface water and groundwater conversion and related literatures. However, based on our recent research work, we have added some detailed information that the mutual transformation of groundwater and surface water in the region.
- In the part ‘3.1.’: There are so many samples (n=156) from different sites, but there seems to be a tendency to generalize too simply that groundwater was affected by anthropogenic activities, such as sewage or chemical fertilizers. A more detailed discussion by region or sample is needed.
Response: We have discussed the characteristics of groundwater according to the different land use kinds in the section 3.2.
- In the part ‘3.1.’: It would be helpful if you could explain the damage caused by the high Fe concentration in the groundwater as mentioned about the effect of high values of TH in groundwater.
Response: We have explained the damage caused by the high Fe concentration in the groundwater in the 3.1 section.
- In the part ‘3.2.’: “(figure 2)” is missing from the first sentence.
Response: Thank you for your reminder. We have added the figure 4 in the part ‘3.3.’
- In the part ‘3.2.’: The authors said that “which show that the groundwater chemistry have been influenced by human activity.” In this sentence, please clarify what kind of human activity it is.
Response: We have added the kind of human activity affecting the groundwater chemistry.
- In the part ‘3.2.’: The authors said that “By contrast, the main hydrochemical types of groundwater in urban and industrial areas were the SO4 and Cl types, reflecting the influence of intense human activity in these areas on the groundwater chemistry.” In this sentence, isn't it Na, not Cl? And please clarify what kind of intense human activity it is.
Response: Thank you for your reminder. We have revised “By contrast, the main hydrochemical types of groundwater in urban and industrial areas were the SO4 and Cl types, reflecting the influence of intense human activity in these areas on the groundwater chemistry.” as “By contrast, the main hydrochemical types of groundwater in urban and industrial areas were the SO4, Na and Cl types, reflecting the influence of intense human activity (such as domestic sewage and industrial wastewater) in these areas on the groundwater chemistry.”.
- In the part ‘3.3.’: In the last sentence, Is there any direct basis or reason for interpreting the 24 samples as human activities?
Response: Thank you for your reminder. There is not direct basis for interpreting the 24 samples as human activities. Thus, we have revised our statement.
- In the part ‘3.4.’: Just because the correlations between TH, TDS, Na+, Ca2+. Mg2+, Cl-, HCO3-, and SO42- are good, how can they all be of the same origin? Please explain more clearly.
Response: This conclusion is based on the statistical knowledge and hydrochemical theory. First, if there is a strong correlation between the two sets of variables, then, they are affected by the same factors. In the aquatic environment, if there is a strong positive correlation between the two indicators, then the concentration of them will show a synchronous increase, which means that they are affected by the same control factors. Therefore, they must come from the same source (the sources could be one or more). This theory has been widely used in the source identification of water and soil environment. For example, references (Hu Y , He K , Sun Z , et al. Quantitative source apportionment of heavy metal(loid)s in the agricultural soils of an industrializing region and associated model uncertainty[J]. Journal of Hazardous Materials, 2020, 391:122244.) and (Ren, C. and Zhang, Q., 2020. Groundwater Chemical Characteristics and Controlling Factors in a Region of Northern China with Intensive Human Activity. International Journal of Environmental Research and Public Health, 17(23): 9126.) In addition, these results were confirmed by principal component analysis in the section 3.7.
- In the part ‘3.4.’: The authors said that “there was a strong positive correlation between COD, Fe, and Mn, indicating their common source.” Please explain more specifically.
Response: The answer to this question is the same as the question 9.
- In the part ‘3.4.’: The authors said that “NO3- and NO2- may have originated from different sources with other hydrochemical parameters.” Please give any example of what other mechanisms are?
Response: The answer to this question is the same as the question 9 and 10. Because the NO3− and NO2− were not significantly correlated with other indicators, which mean they are affected by the different control factors with the other parameters. Thus, we think that the NO3- and NO2- originated from different sources with other hydrochemical parameters. For example, for a river, if the upper river water is mainly affected by rainfall (the content of ion is low), and the lower river water is affected by domestic sewage (the content of ion is high), then, there is no good correlation between the chemical components of the upstream and downstream (the concentration of ions were no synchronous increase or decrease).
- In the part ‘3.5.’: In the third paragraph (Factor 2), the discussion of COD is missing.
Response: We have discussed the sources of COD in this area in the part ‘3.7’.
- It would be better if latitude and longitude information were added to Figure 1.
Response: We have added the information of latitude and longitude in Figure 1.
- Consequently, this paper presented all the importance of natural (e.g. rock weathering) and anthropogenic as influencing (primary) factors for 156 groundwater samples. However, it seems necessary to at least discuss the difference in importance according to local environmental characteristics or geological characteristics or location in depth.
Response: We have discussed the characteristics of groundwater according to the different land use kinds in the section 3.2. In addition, we have discussed the hydrochemical type for the different types of groundwater.
Reviewer 3 Report
I have reviewed Manuscript titled "Assessment of the Evolution of Groundwater Chemistry and its Controlling Factorsin the Huangshui River Basin, of North-Western China, Using Hydrochemistry and Multivariate Statistical Techniques". The material is presented appropriately and clearly, the data contained in figures represent understandable documentation of the research problem, however I suggest a minor revision and propose a few correction (listed below).
- Page 2 „There are five types of groundwater in this area: (1) loose stratum pore water, (2) water in clastic rock fissures, (3) water in karst carbonate, (4) water in bedrock fissures, and (5) frozen water layers.” – show it in geological cross section, it is difficult to interpret the chemistry of water without knowing geology. The description of geology is too general.
- 1 – „glacier” and „construction land” - very similar colors make them unidentifiable on the map. Explain the symbols "G124" and others.
- Fig 2 is not quoted in the text nor discussed.Please add a comment to this figure or remove it, or move to section2.
- Section 3.1. „The CO32-concentrations in the groundwater samples were all below the detection limit.”, however, in Table 1 and the methodology section there is no information that this anion was analyzed.
- Page 6 - How will the Authors explain that "the aquifer is in an overall oxidizing environment" but the groundwater are contaminated "the groundwater has been influenced by anthropogenic activities, such as sewage and chemical fertilizers"? Usually contaminated of groundwater do not contain oxygen (lack of oxygen is an indicator of contamination).
- Since there are five types of groundwater in the study area, each type should be analyzed separately. It is not methodically correct to interpret such different types of water together (differences in the geology of the aquifer, differences in the position of the water table). In my opinion, this suggestion requires some thought and correction.
- Please correct the title of Table 2. "huangshui River basin" to "Huangshui River basin".
- The information presented in Table 3 would be worth presenting on a land use map. Then the impact of land using on the groundwater chemistry would be evident.
- Page 8 – „Therefore, NO3− and NO2− may have originated from different sources with other hydrochemical parameters” – from what sources?
- Page 10 – “Rock weathering was the primary factor influencing the hydrochemistry of groundwater in this region...” - please write what rocks are there.
- General opinion, the interpretation does not refer to the geological structure (too general description) and hydrogeological conditions (only the mentioned position of the water table).These two factors are very important in analyzing the evolution of groundwater.It is worth presenting this more detailed.
Author Response
I have reviewed Manuscript titled "Assessment of the Evolution of Groundwater Chemistry and its Controlling Factorsin the Huangshui River Basin, of North-Western China, Using Hydrochemistry and Multivariate Statistical Techniques". The material is presented appropriately and clearly, the data contained in figures represent understandable documentation of the research problem, however I suggest a minor revision and propose a few correction (listed below).
Response: Thank you for your suggestions.
- Page 2 „There are five types of groundwater in this area: (1) loose stratum pore water, (2) water in clastic rock fissures, (3) water in karst carbonate, (4) water in bedrock fissures, and (5) frozen water layers.” – show it in geological cross section, it is difficult to interpret the chemistry of water without knowing geology. The description of geology is too general.
Response: We have added some geologic description of study area, and added the figure of hydrogeological cross-section in the part ‘2.1’.
- 1 – „glacier” and „construction land” - very similar colors make them unidentifiable on the map. Explain the symbols "G124" and others.
Response: We have changed the color of glacier to yellow. The symbol for site G124 is the same as the other sites in Figure 1.
- Fig 2 is not quoted in the text nor discussed. Please add a comment to this figure or remove it, or move to section2.
Response: Thank you for your reminder. We have quoted the Fig 2 in the 3.2 section.
- Section 3.1. „The CO32-concentrations in the groundwater samples were all below the detection limit.”, however, in Table 1 and the methodology section there is no information that this anion was analyzed.
Response: Thank you for your reminder. We have supplemented the analysis method of CO32- in Table 1.
- Page 6 - How will the Authors explain that "the aquifer is in an overall oxidizing environment" but the groundwater are contaminated "the groundwater has been influenced by anthropogenic activities, such as sewage and chemical fertilizers"? Usually contaminated of groundwater do not contain oxygen (lack of oxygen is an indicator of contamination).
Response: In this study area, the aquifer is in an overall oxidizing environment. This mainly based on the average oxygen content (mean DO values is 5.75mg/L) of all sample (n=156). Indeed, as you said, hypoxia is an indicator of contamination. In our study area, although groundwater is affected by human activity, only 10 samples have DO concentrations below 2 mg/L. This is mainly due to that the contaminated components (such as Na+, SO42-, TH and Fe) are mainly inorganic components, and their consumption of oxygen is not too serious. Moreover, our sampling sites are mainly located in the valley area (on either side of the river), and the recharge of surface water to the groundwater is relatively strong, thus, this in turn increases the content of DO in the groundwater.
- Since there are five types of groundwater in the study area, each type should be analyzed separately. It is not methodically correct to interpret such different types of water together (differences in the geology of the aquifer, differences in the position of the water table). In my opinion, this suggestion requires some thought and correction.
Response: Thank you for your suggestions. In this study, we mainly collect two type of groundwater because they were mainly used by the local residents. We have added the discussion of hydrochemical types of two type groundwater in the section 3.2.
- Please correct the title of Table 2. "huangshui River basin" to "Huangshui River basin".
Response: Thank you for your reminder. We have revised "huangshui River basin" as "Huangshui River basin".
- The information presented in Table 3 would be worth presenting on a land use map. Then the impact of land using on the groundwater chemistry would be evident.
Response: Thank you for your suggestions. We have added the hydrochemical type of the sampling site to the land use type map(Fig.5).
- Page 8 – „Therefore, NO3− and NO2− may have originated from different sources with other hydrochemical parameters” – from what sources?
Response: We have added the source of the NO3− and NO2−. However, the specific source is referred to the section 3.7. This section focuses on discussing the correlation among the groundwater chemical parameters, and the origin of hydrochemical components was briefly discussed. In the section 3.7, the source of each ion will be analyzed in depth.
- Page 10 – “Rock weathering was the primary factor influencing the hydrochemistry of groundwater in this region...” - please write what rocks are there.
Response: We have discussed the rock weathering in the section 3.5 and identified that the silicate rock weathering is the main type of rock weathering
- General opinion, the interpretation does not refer to the geological structure (too general description) and hydrogeological conditions (only the mentioned position of the water table).These two factors are very important in analyzing the evolution of groundwater. It is worth presenting this more detailed.
Response: We have added the description of the geological structure and the figure of hydrogeological cross-section in the section‘2.1’. But, in this study area, the research degree of hydrogeology is relatively low. Therefore, we didn't find any more information about the geological structure. In addition, we have added the hydrogeochemical graphs to evaluate the groundwater evolution in the section 3.4 and discussed the hydrochemical type for the different types of groundwater in the section 3.3. There information can provide support for accurately identifing the evolution of groundwater.
Round 2
Reviewer 1 Report
I agree with the modifications made to the manuscriptReview punctuation marks
Author Response
Comments and Suggestions for Authors
I agree with the modifications made to the manuscript
Review punctuation marks
Response: Thank you for your suggestions. We have revised the punctuation marks of paper and edited the English language and style by a native English speaker.